# New Evidence of Tiger Subspecies Differentiation and Environmental Adaptation: Comparison of the Whole Genomes of the Amur Tiger and the South China Tiger

**DOI:** 10.3390/ani12141817

**Published:** 2022-07-16

**Authors:** Hairong Du, Jingjing Yu, Qian Li, Minghai Zhang

**Affiliations:** 1College of Wildlife and Protected Area, Northeast Forestry University, Harbin 150040, China; dhr9012@163.com (H.D.); jinjinyu1987@163.com (J.Y.); 2Resources & Environment College, Tibet Agricultural and Animal Husbandry University, Nyingchi 860000, China; 3College of Pharmacy, Guizhou University of Traditional Chinese Medicine, Guiyang 550025, China

**Keywords:** Amur tiger, South China tiger, whole-genome sequencing, environmental adaptation, genetic evolution

## Abstract

**Simple Summary:**

Tigers are top predators and umbrella protectors, vital to the stability of ecosystems. The South China tiger has been declared extinct in the wild and only exists in captivity. The Chinese government is actively promoting the reintroduction of the South China tiger into the wild. The future of the wild population of the Amur tiger in China is not optimistic, and the recovery of the population is an essential task for the conservation of the Amur tiger. The recovery of the population is not only a macroscopic problem but also a significant study of molecular ecology. We used high-throughput sequencing technology to study the differences in adaptive selection between Amur tigers and South China tigers. Significant genetic differences were found between the Amur tiger and the South China tiger based on a principal component analysis and phylogenetic tree. We identified functional genes and regulatory pathways related to reproduction, disease, predation, and metabolism and characterized functional genes related to survival in the wild, such as smell, vision, muscle, and predatory ability. The data also provide new evidence for the adaptation of Amur tigers to cold environments. *PRKG1* is involved in temperature regulation in a cold climate. *FOXO1* and *TPM4* regulate body temperature to keep it constant. The research also provides a molecular basis for future tiger conservation.

**Abstract:**

*Panthera tigris* is a top predator that maintains the integrity of forest ecosystems and is an integral part of biodiversity. No more than 400 Amur tigers (*P. t. altaica*) are left in the wild, whereas the South China tiger (*P. t. amoyensis*) is thought to be extinct in the wild, and molecular biology has been widely used in conservation and management. In this study, the genetic information of Amur tigers and South China tigers was studied by whole-genome sequencing (WGS). A total of 647 Gb of high-quality clean data was obtained. There were 6.3 million high-quality single-nucleotide polymorphisms (SNPs), among which most (66.3%) were located in intergenic regions, with an average of 31.72% located in coding sequences. There were 1.73 million insertion-deletions (InDels), among which there were 2438 InDels (0.10%) in the coding region, and 270 thousand copy number variations (CNVs). Significant genetic differences were found between the Amur tiger and the South China tiger based on a principal component analysis and phylogenetic tree. The linkage disequilibrium analysis showed that the linkage disequilibrium attenuation distance of the South China tiger and the Amur tiger was almost the same, whereas the r^2^ of the South China tiger was 0.6, and the r^2^ of the Amur tiger was 0.4. We identified functional genes and regulatory pathways related to reproduction, disease, predation, and metabolism and characterized functional genes related to survival in the wild, such as smell, vision, muscle, and predatory ability. The data also provide new evidence for the adaptation of Amur tigers to cold environments. *PRKG1* is involved in temperature regulation in a cold climate. *FOXO1* and *TPM4* regulate body temperature to keep it constant. Our results can provide genetic support for precise interspecies conservation and management planning in the future.

## 1. Introduction

Rapid human population growth, environmental change, and habitat fragmentation pose ever-greater biodiversity threats and highlight the need for increasingly aggressive conservation efforts (e.g., focused on *Panthera tigris*). The tiger is the largest cat [1] and one of the most endangered species in the world. Currently, the world’s wild individual tigers number from about 2154 to 3159, and most of them are isolated in small populations [2]. The tiger is the key species and natural indicator of the health of ecological communities [3]. Historically, the tiger consisted of nine genetically validated subspecies [4,5], but only five subspecies (Amur, Bengal, Indochinese, Malayan, and Sumatran tigers) remain now [5]. In April of 2007, the Chinese government announced that there were no traces of South China tiger (*P. t. amoyensis*) activity in the wild [6]. Although144 individuals are in captivity [7], their ancestors are six individuals captured from the field. Inbreeding, low genetic diversity, minimal populations, and endangerment currently block South China tiger populations from recovery. Appropriate artificial interference may preserve the population, although low genetic diversity makes it difficult to restore the minimal populations and release them into the wild [8].

Molecular biology technology has been widely used in biological protection, especially the protection of endangered animals and plants [9,10,11,12]. Whole-genome sequencing (WGS) provides insights into genetic information recorded in gene fragments, such as historical population dynamics of species, evolutionary bottlenecks, hybridization, gene penetration, genetic diversity, inbreeding in small populations, etc. Population recovery is an effective way to save endangered species. Currently, the South China tiger is considered extinct in the wild. Only 204 individuals (by the end of 2019) exist in ex situ conservation, all of which were bred from the initial six individuals. [13]. Although the population of captive South China tigers has increased [13,14], inbreeding is inevitable due to the small initial population [8,15]. The dangers of inbreeding are apparent, especially for the South China tiger, which is extinct in the wild [16]. Meanwhile, the South China tiger’s importance as a top predator in the ecosystem is so great that the Chinese government is actively promoting its reintroduction into the wild [17]. Genomic tools facilitate the management of captive populations and the selection of individuals to be released by providing accurate genetic parameters [18].

The future of the wild population of the Amur tiger (*P. t. altaica*) is not optimistic, and population recovery is also an important issue currently facing these tigers. The genome study offers practical molecular biological support for the population of Amur tigers. We consider the most successful case of endangered species recovery to be the genetic recovery of the endangered Florida panther (*Puma concolor coryi*) population [19]. We hope that the recovery of the South China tiger population can be guided by gene rescue, which is based on the screening of the genome. Therefore, a genome-wide study is significant for the conservation management and population recovery of Amur and South China tigers.

The genomes of the Amur tiger and the South China tiger were studied by using whole-genome sequencing technology. We explored: (1) How did the chronological sequence of the Amur tigers and South China tigers evolve? (2) Which functional genes are associated with tiger survival in the wild and can be found by studying functional genes and regulatory channels? The results can provide a scientific reference for the molecular conservation and management of captive Amur tigers and South China tigers in the future and lay the foundation of genetic information for population recovery by providing insights for releasing the selection of individual tigers into the wild.

## 2. Materials and Methods

### 2.1. Sample Collection

In this study, a total of seven tissue samples and one blood sample were collected for whole-genome sequencing. The Amur tiger samples were donated from the Northeast Tiger Park in northeastern Heilongjiang province and the South China tigers were donated from Shanghai Zoo. The experimental samples were placed in the refrigerator at −20 °C after collection. The detailed information of the eight samples is shown in Appendix A.

### 2.2. DNA Extraction and Construction of a Genome Library

Tissue and blood samples were collected, and the DNA of each sample was extracted from tissue samples using standard phenol–chloroform methods. The steps were as follows: (1) ground the tissue samples to powder; (2) dissolve the tissue powder with STE, then add protease K to obtain the final concentration of 100 μg/mL; (3) shake in a 55 °C water bath overnight for digestion; (4) add the same volume of phenol and extract 1 time, then follow by shaking for 5 min and centrifugation at 12,000× *g* for 5 min; (5) take the supernatant and add phenol:chloroform:isopentyl alcohol (25:24:1), extract once, shake slightly for 5 min, and centrifuge 12,000× *g* for 5 min; (6) take the supernatant and add chloroform:isoamyl alcohol (24:1) to extract once and centrifuge; (7) take the supernatant, add iced ethanol two times and 1/10 of 3M sodium acetate, thoroughly mix 12,000× *g* by centrifugation for 5 min, then remove the supernatant; (8) rinse twice with 500μg of 75% ethanol; (9) finally, dissolve dd H20. Blood samples were extracted using a QIAamp DNA Mini Kit (Qiagen, https://www.qiagen.com/cn/shop/automated-solutions/qiaamp-dna-blood-mini-kit) (accessed on 1 August 2020).

The qualified DNA samples were randomly interrupted by a Covaris ultrasonic crusher, the fragment ends were repaired, and a poly(A) tail was added. Sequencing joints were connected, purification, PCR amplification, and filtering were completed, and the whole library was prepared. After the library was qualified, Illumina HiSeq/MiSeq sequencing was performed by pooling different libraries according to the requirements of effective concentration and the target data amount. The constructed library was double-ended by an Illumina Hiseq PE150 sequencer, and the whole genome of reads with a length of 150 bp was sequenced. The sequencing depth was targeted at an average of >25×, and the average coverage was 99. 82% (coverage of at least one base), producing an average of 77.01 Gb of raw sequencing data per individual.

### 2.3. Resequencing and Variant Identification

Total genomic DNA from 8 tigers was extracted as described by Diversity Arrays Technology for Illumina sequencing. Paired-end sequencing libraries with insert sizes of 150 bp were constructed according to the manufacturer’s instructions for sequencing on the HiSeq PE150 platform. Paired-end reads (clean reads) obtained from sequencing were mapped to the Amur tiger genome [20] using the MEM algorithm from the BWA software [21]. The parameters used were “mem -t 4 -k 32 -M”.

SAMtools [22] was used to convert mapping results to bam format. The bam alignments were then converted to pileup and glf formats using the pileup command. SNPs and InDels were then detected using BCFtools; the Bayesian model was used to determine the base’s mutation type and polymorphism loci. Sites with a probability to be a variant >0.99 were further extracted to identify the putative SNP based on the following criteria: copy number <1.5, sequencing depth according to an average depth of each accession, 1000 < sequencing depth <3000, and SNPs a minimum of 5 bp apart, except for minor allele frequencies (MAF >0.05). ANNOVAR [23] was used to calculate the likelihood of genotypes of each individual. In each individual, SNPs were filtered by the quality value (>20), the minimum number of required reads supporting each SNP (>4), the maximum overall depth (<100), the maximum copy number of flanking sequences (<1.5), and the *p*-value of the rank sum test (*p* > 0.05).

### 2.4. Copy Number Variation

Copy number variations (CNVs), which include deletions, duplications, and large-scale copy number variants or copy-number polymorphisms as well as insertions, inversions, and translocations, are believed to contribute significantly to variations between individuals and may have as large an effect on phenotype as SNPs [24]. CNVnator [25] (parameters “-call 100”) was used to detect and screen to obtain the number, structure, and location of CNVs. Read depth per 100 bp window was computed using this modified software, which adjusts for bias in reading depth caused by GC content, and mutation annotation was performed through the ANNOVAR (ANNOtate VARiation) tool [23,26].

### 2.5. Calculation of LD

Correlation coefficient values (r^2^) of alleles were calculated using Haploview to measure the LD level in the two populations. The parameters were set as follows: “-dprime -max Distance 1000 -min MAF 0.05 -hwcutoff 0.001 -missing Cutoff 0.5 -min Geno 0.6”. The mean r^2^ was adopted to represent the average LD for each group and LD decay figures were drawn using R [27].

### 2.6. Selective Sweep Analysis

The population of genetic summary statistics (θπ, θw, Tajima’s D and Fst; Appendix A) was calculated for the two populations. The following criteria were used to identify candidate genes in each of the two pairwise comparisons: Fst values >95% of the population pairwise distribution; θπ and θw higher in A and < 10% of the B population distribution; negative Tajima’s D values in the A/B population. The mean log-likelihood-ratio test statistic assessed significance. The heterozygosity log_2_(θπH/θπD ratio) was calculated over a 100 kb sliding window with a step of 20 kb. All annotated genes that overlapped with sweep windows or their flanking windows (20 kb up- and downstream of the sweep region) were defined as candidate genes. GO categories [28] assessed gene family enrichment across subsets of genes under selection and invariant genes. The analysis first mapped all the candidate genes to the related term in the Gene Ontology database (http://www.geneontology.org/ accessed on 13 August 2020), calculated the number of target genes mapped to each term, and then used hypergeometric tests to identify the GO entries that were significantly enriched in the candidate target genes compared with the whole reference genome, in which the *p* values were corrected using the Benjamini–Hochberg method [29].

## 3. Results

### 3.1. Sequencing and Variation

This study selected eight tigers with two distinct geographic distributions, including five Amur tigers and three South China tigers (Figure 1a). We sequenced these tigers to an average of >25× genome coverage depth using the Illumina HiSeq PE150 to generate 150 bp paired-end reads. The raw reads were subjected to a series of quality control procedures (Appendix A), and then the filtered high-quality reads were mapped back to the Amur tiger genome (GCA_000464555.1).

Resequencing of the tiger yielded 648 billion 150 bp paired-end reads, which comprised 647 Gb of high-quality raw data (Appendix A). Sequence reads were aligned to the Amur tiger reference genome (SRA074975) using BWA software [21]. The mapping rate in different individuals varied from 97.55% to 98.16%, averaging 97.96%. The average final effective mapping depth achieved was ~28× per individual, ranging from 25× to 30×.

Using a conservative quality filter pipeline (see Materials and Methods Section 2.3), we identified 6,338,767 SNPs genome-wide in two subspecies (Appendix A) based on comparisons to the reference genome. Of the 6.3 million high-quality SNPs, most (66.3%) were located in intergenic regions, with an average of 31.72% located in coding sequences (Appendix A; Figure 1b,c). Coding regions displayed lower diversity levels relative to intron and UTR sequences. There were 25,148 synonymous and 18,534 non-synonymous SNPs among the coding areas, resulting in a non-synonymous-to-synonymous substitution ratio of 0.74. Moreover, we investigated the SNPs that are likely to significantly impact gene function, including mutations that lead to stop gain, stop loss, and altered splicing.

In addition to SNPs, we identified 1,727,980 small insertions and deletions (InDels) in the Amur tiger and South China tiger. The InDels ranged from 1–200 bp in length (Appendix A). Most InDels (99.58%) were small (1–6 bp), with only 0.42% greater than 20 bp in length (Figure 1d; Appendix A). The majority (64.76%) of the InDels were located in intergenic regions, and 32.34% were located in coding sequences, among which only 0.38% were in-frame, three bp InDels. These genes with large effect mutations might be important in the functional evolution of tiger genes. This dataset provides a set of molecular markers that could be used to identify evolutionary traits in related genes [1,30,31,32].

268,187 copy number variations (CNVs) were identified (Figure 1e; Appendix A). The number of SNPs was higher in the Amur tiger than in the South China tiger (Appendix A). As a result, the sample sizes of the two subspecies are not equal. After the number of individuals was normalized, there were about 780,000 SNPs in the South China tigers, and about 1.86 million in the Amur tigers, which is around 2.4 times as much as the former. Therefore, it can be speculated that the South China tiger has lower polymorphism. Since the South China tiger is extinct in the wild, all of the captive South China tigers are the offspring of six individuals captured in the mid-20th century. This led to a higher degree of inbreeding, decreasing genetic heterogeneity and genetic diversity.

### 3.2. The Divergence between the Amur Tiger and the South China Tiger

Recent studies [1,30,31,32] proposed that Amur tigers came from the South China tiger. To observe the divergence between the Amur tiger and the South China tiger at the genomic level, we constructed a rooted neighbor-joining phylogenetic tree based on 6,338,767 high-quality SNPs (Figure 2a). The Amur tiger and the South China tiger were separated into two large clades. In addition, we used mitochondrial genomes to construct phylogenetic trees for seven tiger subspecies and Felis catus, which were divided into two large clades. The phylogenetic tree produced from eight species proves that the Amur tiger and the Bengal tiger are all differentiated from the South China tiger (Figure 2b), which is consistent with the findings of Junsong Shi [32].

The principal component analysis (Figure 2c) shows that the South China tiger separated on main component 1, followed by the Amur tiger on main component 2. We performed population structure analysis using the software PLINK, with K changing progressively from 2 to 5 and dividing the tigers into two subspecies. The results of the population structure analysis were similar to that of the evolutionary tree and principal component analysis, as can be seen from Figure 2d; when K = 2, it was evident that the eight individuals were divided into two groups, which was consistent with the basic information of the sample collected in this experiment. There was a small gene exchange between the Amur tiger and the South China tiger. When K = 3, five Amur tigers were divided into four groups, which was consistent with the genetic distance of two Amur tigers in the principal component analysis. When K = 4, the five Amur tigers were distinct. When K = 5, the three South China tigers were also separate. This shows that there is a specific genetic distance between each individual. The small gene exchange means that both do not have complete geographical isolation. All the above analyses consistently showed that they have a closer phylogenetic relationship.

We estimated the population diversity parameters, π and θω, and found that the overall nucleotide diversity in the Amur tiger was higher than that in the South China tiger (Table 1). The linkage disequilibrium (LD) distance of the two subspecies was almost similar (between 54.5 kb and 48.3 kb) (Figure 2e). The LD was increased markedly in the South China tiger compared with the Amur tiger, with the LD reaching half of its maximum value when the r^2^ value of the Amur tiger is about 0.4 and the South China tiger is about 0.6. The South China tiger population has a high degree of LD compared to the Amur tiger population.

### 3.3. Screening and Annotation of Selective Sweeps based on SNP

We explored the genomic regions with high divergence to identify essential genes that possibly govern physiological traits, such as environmental adaptions. We calculated the fixation index (F_st_) [33] between subspecies, and the genomic regions with F_st_ values in the top 5% were considered highly differentiated (Figure 3a; Appendix A).

We performed a selective sweep analysis over the whole genome to identify candidate genes under positive selection in different tiger subpopulations. We scanned the genome in 40 kb sliding windows with a step size of 20 kb. We calculated the reduction of diversity based on the population differentiation F_st_ and π values (Appendix A) in the Amur tiger and the South China tiger, respectively (Figure 3b–d). The regions in the top 5% were considered as selective sweeps. Fisher’s exact test obtained the significance of all statistics. In total, we identified 702 and 414 candidate genes in the Amur tiger and the South China tiger, respectively (Appendix A).

We performed a Gene Ontology (GO) enrichment analysis of the gene sets in sweep regions of the Amur tiger and the South China tiger. We found that genes with essential biological functions, such as reproduction, carnivorous diet, metabolism, and disease were significantly enriched in the Amur tiger (Appendix A). For example, a set of key terms was found related to the process of mating and reproduction, such as spermatid development (GO:0007286), mating-type factor pheromone receptor activity (GO:0004932), and hormone activity (GO:0005179). Moreover, other enriched functions include regulation of response to stress (GO:0080134), regulation of defense response (GO:0031347), immune response (GO:0006955), cardiovascular system development (GO:0072358), as well as many GO clusters in the regulation of blood pressure vasodilation (GO:0008217, GO:0042311). All functional terms mentioned above had a significant enrichment score (*p*-value < 0.05) after considering multiple testing errors using the Bonferroni method. Besides, categories associated with muscle function, such as smooth muscle contraction (GO:0006939), muscle organ development (GO:0007517) and muscle structure development (GO:0061061), and sensory organs (GO:0050909, GO:0007600, GO:0007601, GO:0007608) were also characterized. Then, we performed *KEGG* pathway analysis on the genes in these regions, and 220 pathways were obtained; some *KEGG* pathways are shown in Appendix A. These pathways are mainly metabolized, e.g., glycerolipid metabolism, tyrosine metabolism, nitrogen metabolism, tryptophan metabolism, pyruvate metabolism, and glycerophospholipid metabolism. These amino acid metabolic signals are associated with a compulsive carnivorous diet. Signal transduction occurs in many pathways, e.g., *HIF-1* signaling pathway, *cGMP-PKG* signaling pathway, *cAMP* signaling pathway, *TNF* signaling pathway, calcium signaling pathway, insulin signaling pathway. These pathways are related to sight and smell, which are important for the ownership of territory, mating, and hunting. Synthesis of hormones includes N-Glycan biosynthesis, fatty acid biosynthesis, steroid hormone biosynthesis, and primary bile acid biosynthesis. The diseases include legionellosis, toxoplasmosis, pertussis, leishmaniasis, dilated cardiomyopathy, and malaria. The significant enrichment pathway (*p* < 0.05) is shown in Table 2.

Through studying the selected regions, we found that both tigers have a lot in common with regard to the enrichment of gene function and metabolic pathway analysis, such as in the carnivorous diet, muscular strength, metabolic diseases, etc. Tigers are located at the top of the food chain in mammals, thus they have high vigilance, high-intensity energy metabolism, and muscle strength. Additionally, their keen senses of smell, hearing, and sharp instincts alone are enough to make them adapt to a complex ecological environment, which corresponds to their rank in the niche.

In the two subpopulations, we observed a consistently strong signal of positive selection on chromosome 10 NW_006712238.1, which harbors a Toll-like receptor 4 (*TLR4*) gene (Table 3). The *TLR4* gene was found to be involved in many metabolic pathways related to legionellosis, toxoplasmosis, malaria, amoebiasis, and leishmaniasis. The protein encoded by the *TLR4* gene is a member of the Toll-like receptor (*TLR*) family, which plays a fundamental role in pathogen identification and innate immune activation. Its activation leads to cell signaling pathways within the NF-κB and the production of inflammatory cytokines, which activate the innate immune system [34]. Genetic diagnosis can be used to study the mechanism of the parasite and perform early prevention and timely diagnosis.

In addition to *TLR4*, we also characterized some of the two subspecies’ candidate genes with a significantly high window Fst and heterozygosity ratio (HP) that are functionally plausible for adaptation in the wild. For example, these genes include: (1) *TEX22*, *FOXO1*, *FBXL17*, *MCTP1*, and *ASB2* associated with smell; (2) *FAM189A1* and *PDE6H* associated with vision [35,36]; (3) *ADAMTSL3*, *FAM189A1*, *FAM135B*, *DPP10* and *FBXL13* involved in muscle development [37]; and (4) *EFHC2*, *CDH13*, *DNM3*, *RAD51D*, *CDH13*, *GPC6*, *ROBO1*, *LECT2* and *PLB1* for regulating production [23,38,39,40,41,42,43,44].

### 3.4. Selective Elimination Analysis based on CNV

The analysis of CNV in the selected region of the tiger’s genome was identified by selecting the signal. Figure 3e shows the distribution of V_st_ of the Amur tiger population and the South China tiger population in the genome (0 ≤ Vst ≤ 1, the closer the V_st_ approaches 1, the more obvious the differentiation is between subpopulations; Appendix A). The V_st_ threshold in the 5% selected area is 0.493. Accordingly, 782 selected regions were identified in this study, with a total length of 3.9 Mb and an average length of about 4.9 kb, containing 94 genes (Appendix A). As can be seen from Figure 3e, there are more outlier sites between the Amur tiger and the South China tiger. Many chromosome regions have large V_st_ values, and the largest V_st_ value is 0.999, which is located in the chromosome NW_006712148.1 region, and there are seven CNVs in the selected region.

To assess possible gene functions targeted by both the Amur tiger and the South China tiger, we performed a Gene Ontology (GO) enrichment analysis of the gene sets in sweep regions of the two subpopulations (Appendix A). The results showed many irritability-related functional terms over-represented in their overlapping candidate genes (Appendix A). For example, a set of key terms was found related to stimulus–response, such as cellular response (GO:0051716, GO:0051716). Categories associated with signal receptor activity, such as signal transduction (GO:0007165), G-protein-coupled receptor signaling (GO:0007186), and cell communication (GO:0007154). Moreover, other enriched functions include biological regulation (GO:0065007), integral to membrane functions (GO:0016021), and receptor activity (GO:0004872). All functional terms mentioned above had a significant enrichment score (*p* < 0.05) after considering multiple testing errors (by the hypergeometric test/Fisher’s exact test).

The genes in these selected regions were analyzed via the *KEGG* pathway, and 63 biochemical pathways were obtained (Appendix A). The significant enrichment (*p* < 0.05) pathway is olfactory transduction. In addition, these pathways are mainly metabolized, as found with phenylpropane metabolism, phenylalanine metabolism, β-alanine metabolism, tyrosine metabolism, retinol metabolism, and olfactory transduction. Interestingly, some disease pathways were also found, such as autoimmune thyroid disease, type I diabetes mellitus, type II diabetes mellitus, Epstein–Barr virus infection, legionellosis, viral myocarditis, toxoplasmosis, measles, systemic lupus erythematosus, influenza A, Huntington’s disease, *HTLV-I* infection, and so on. These are consistent with the results of the *KEGG* metabolic pathway based on SNP.

### 3.5. The Differentiation Mechanism of Tiger Subspecies

Bergmann’s rule states that endothermic animal subspecies living in colder climates have larger bodies than that of the subspecies living in warmer climates [45]. Individuals with larger bodies are better suited for colder climates because larger bodies produce more heat due to having more cells, and have a smaller surface area compared to smaller individuals, which reduces heat loss. Compared with the South China tiger, the Amur tiger is relatively large in body size, longer in fur, thinner in pattern, and lighter in color. Additionally, it has a very thick white coat around its neck to accommodate the cold weather; our data also provided new evidence of how the Amur tiger adapts to the cold. By comparing our candidate gene list with those from the Amur tiger, we found *PRKG1* as a consistent signal in response to cold adaptation identified among independent approaches. Our results indicated that *PRKG1* was under positive selection in Amur tiger populations. The *PRKG1* gene can be involved in smooth muscle contraction [46,47] and the maintenance of normal blood pressure [48,49], which can help to avoid heat loss and is also important for body temperature regulation. The *PRKG1* protein is best known for its cardiovascular and neuronal functions and it is expressed in cerebellar Purkinje cells, hippocampal neurons, and the lateral amygdala, as well as platelets. In mammals, *PRKG1*-phosphorylated proteins are known to regulate cardiac function, gene expression, feedback of the NO-signaling pathway, and processes in the central nervous system including axon guidance, hippocampal and cerebellar learning, circadian rhythm, and nociception [50].

Nevertheless, it is intriguing that other possible candidate genes of cold adaptations, such as *FOXO1* and *TPM4*, more often exhibit lineage-specific signals. The mammalian body always works to remain in homeostasis. One form of homeostasis is thermoregulation. The normal temperature of the tiger is between 37.5 °C and 39.5 °C. Stress from extreme external temperatures can cause the body to shut down [51]. *FOXO* proteins are major targets of insulin action, and *FOXO1* mediates the effects of insulin on hepatic glucose metabolism [52]. *FOXO1* is significantly expressed in the gene of the Amur tiger and regulates insulin secretion, which acts on carbohydrate metabolism and, thus, increases energy production [53]. Therefore, we believe that *FOXO1* is closely related to body temperature maintenance. These cold temperatures commonly result in mortality. Tigers have adapted to living in these extreme climates primarily through autoregulation. Evidence suggests that tigers may have lived in the ice age and then spread to Africa and the Americas, suggesting that the tigers were adaptable enough to live in various climatic conditions early on. This is supported in the variability selection hypothesis, which says that mammalian adaptability came from environmental change over the long term [54,55].

## 4. Discussion

The evolution of the tiger is known from several pieces of evidence of fossil and phylogenetic analysis. The oldest fossil, called *Felis palaeosinensis*, was found in northern China and Java [56]. It is thought to date from the end of the Pliocene and the beginning of the Pleistocene, being more than two million years old [57]. Collier and O’Brien supported this hypothesis by showing that the tiger diverged from other *Panthera* species up to two million years ago, before the divergence of the lion, leopard, and jaguar [58]. The center of evolution is thought to be in northern China [59]. The Amur tiger and the South China tiger belong to the genus panther in the cat family. Both have the same adaptive ability, which is consistent with the carnivorous and muscular strength of the carnivorous panthers. The Amur tiger and the South China tiger genomes show similar repeat compositions and high genomic synteny, indicating strong genomic conservations in Felidae [20].

Geographical isolation promotes the differentiation of species or subspecies. Tigers in the Russian far east are larger than those in Sudan Island, which may be due to growth hormones stimulated by cold weather, and may also be responsible for the divergence between the South China tigers and the Amur tigers. To adapt to the cold climate, the species can minimize the loss of unit energy by increasing the surface area, as the larger weight can store more energy and fat to resist the cold. Long-term adaptation can lead to some genes being fixed in this trait. Selective elimination occurs when a rare or previously nonexistent allele (relative to individuals of other populations) increases rapidly under the action of natural selection. Through gene selection analysis, we found that *PRKG1* in the genome of the Amur tiger was involved in temperature regulation in a cold environment. *FOXO1* and *TPM4* regulate body temperature to keep it constant, providing new insights for how Amur tigers adapt to cold climates. The body-temperature-related genes *FOXO1* and *TPM4* appear to have been strongly selected in the Amur tiger. Amur tigers live in northeast China and the far east of Russia, where in winter it becomes covered in snow and the average temperature is below −20 °C. The cold environment stimulates the expression of genes related to body temperature regulation in Amur tigers. Recent studies have also shown that there are genes adapted to the cold environment in the Amur tiger genes [60]. The expression of positive selection genes is closely related to energy metabolism and muscle contraction, which may be closely associated with the habitat environment of the Amur tiger. Adaptation to a cold climate may be one of the reasons for the differentiation between the Amur tigers and the South China tigers, similar to the differentiation between the Polar bear (*Ursus maritimus*) and the Brown bear (*U. arctos*) [61].

The LD of the South China tiger was significantly higher than that of the Amur tiger, which may be related to inbreeding and the effective population size of the South China tiger. About 66 years ago, the effective population of the South China tiger was lost [15]. Inbreeding has taken place since captivity in 1963; continuous inbreeding has also led to some problems that are not conducive to population reproduction [8,15,62]. However, the inbreeding of Amur tigers in captivity is low [63]. Principal component analysis and population structure analysis also showed that the South China tiger samples were closely related, whereas the Amur tiger samples were distantly related. Of course, the error caused by the small sample size cannot be ruled out, but this result should also be paid attention to by managers because closely related groups are likely to lead to inbreeding.

The research shows that scientific management could reduce the risk of inbreeding [62]. However, recent studies have claimed that scientific management can only solve the current inbreeding and cannot solve the long-term problem [15]; therefore, the scientists suggest that gene rescue should be used to reduce South China tigers’ inbreeding [8]. Our study found genetic differences in environmental adaptation between the Amur tiger and the South China tiger. The Amur tiger is genetically adapted to the cold environment, which is contrary to the warm environment suitable for the South China tiger. These genes might become novel genetic loads for the South China tiger gene pool given they are introduced. We suggest that the recovery of the South China tiger population using gene rescue should be carefully selected for homologous species, especially for environmental adaptation.

## 5. Conclusions

We used whole-genome sequencing technology to compare and analyze the Amur tiger and South China tiger genomes. The results showed that: 1. The Amur tiger and the South China tiger are separated into two large clades. 2. The South China tiger population has a high degree of LD compared to the Amur tiger population. 3. We identified functional genes and regulatory pathways related to reproduction, disease, predation, and metabolism and characterized functional genes related to survival in the wild. *PRKG1* is involved in temperature regulation in a cold climate. *FOXO1* and *TPM4* regulate body temperature to keep it constant.

## Figures and Tables

**Figure 1 animals-12-01817-f001:**
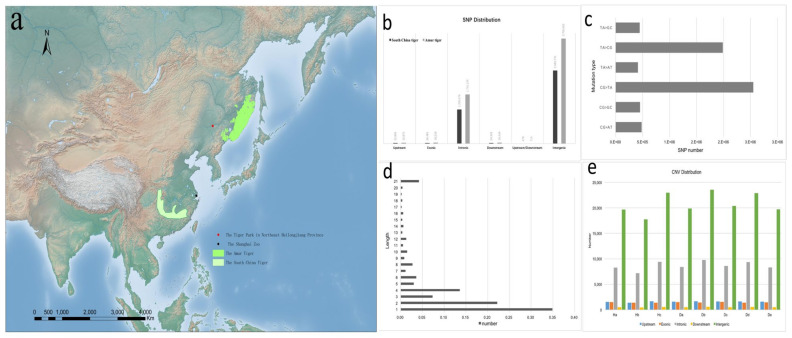
Distribution and variation analysis of tigers in China. (**a**) Distribution of tigers (the distribution of the South China tiger was the last result of a survey by the National Forestry and Grassland Administration; the South China tiger is now considered extinct in the wild). (**b**) SNP location and distribution. (**c**) Mutation type and quantity distribution of SNP. (**d**) InDels length distribution in genome. (**e**) CNV location and distribution.

**Figure 2 animals-12-01817-f002:**
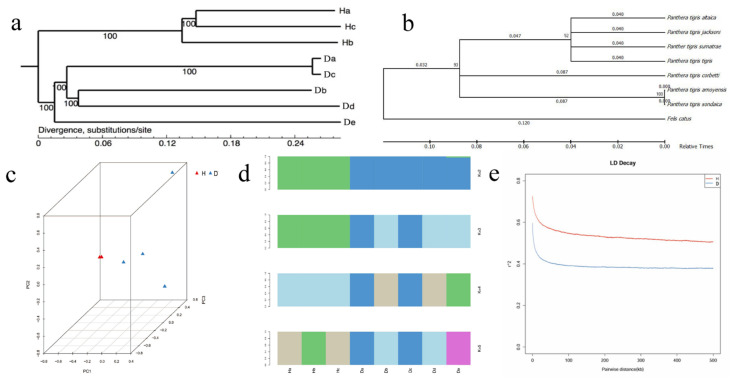
Divergence between the Amur tiger and the South China tiger. (**a**) Phylogenetic relationship analysis of eight tiger breeds. (**b**) Phylogenetic relationship among the *Panthera tigris*. (**c**) Three-dimensional principal component analysis. (**d**) Population structure. Each column in the picture represents an individual, and the length of the different color fragments represents the proportion of an ancestor in the individual genome. (**e**) Decay of linkage disequilibrium (H: the South China tiger; D: the Amur tiger.).

**Figure 3 animals-12-01817-f003:**
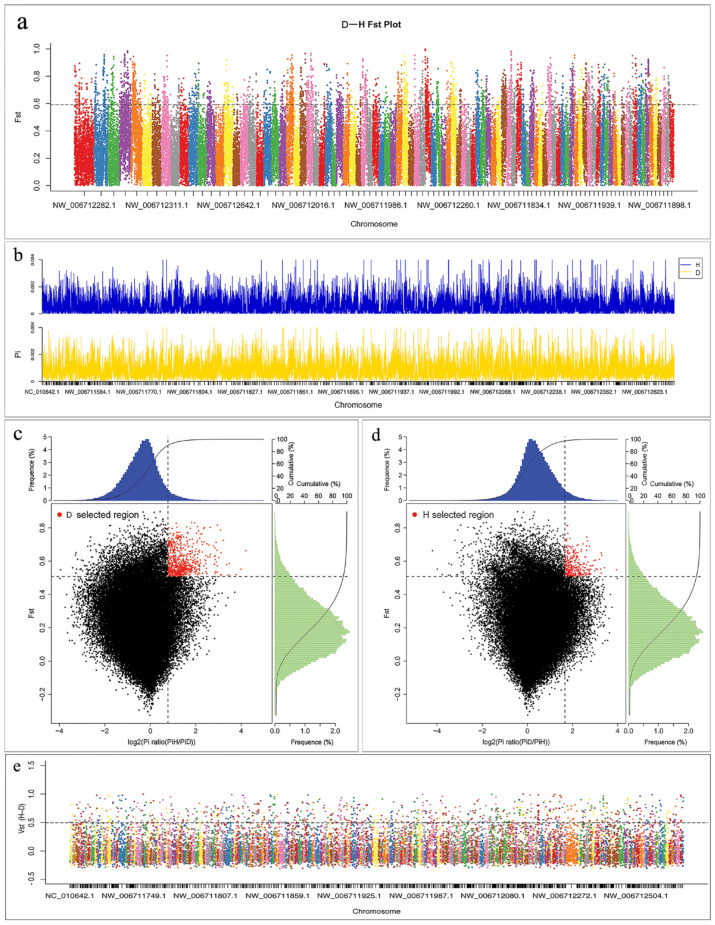
Screening of selective sweeps. (**a**) Genome–wide distribution of F_st_ between the Amur and the South China tiger base on SNP. The horizontal dashed line indicates the threshold defining the selective sweeps (F_st_ ≥ 0.59). (**b**) The θπ distribution. (**c**,**d**) Identification of genomic regions with strong selective sweep signals in Amur tiger and South China tiger. Data points located to the right of the vertical lines (corresponding to 5% right tail of the empirical log_2_(θπ ratio) distribution) and above the horizontal line (5% right tail of the empirical F_st_ distribution) where identified as selected regions (red points). (**e**) Selection analyses identified the selection signal based on V_st_ method. A dashed horizontal line indicates the cut−off (V_st_ > 0.439) used for extracting outliers.

**Table 1 animals-12-01817-t001:** Summary SNP statistics in *Panthera tigris* genotypes.

Genome-Wide	SNP no.	θπ (10–3)	θω (10–3)	Tajima’s D	Non-Syn SNPs	Syn SNPs	Non-Syn/Syn
*Panthera tigris altaica*	5,579,192	0.876	0.812	0.423	16,099	21,948	0.734
*Panthera tigris amoyensis*	3,885,781	0.563	0.526	0.336	11,106	15,268	0.727

**Table 2 animals-12-01817-t002:** KEGG pathway in Amur tiger (*p* < 0.05).

Species	Term	Database	ID	Gene Amount	*p*-Value
Amur tiger	Ribosome	KEGG PATHWAY	fca03010	18	0.0287

**Table 3 animals-12-01817-t003:** KEGG pathway and genes related to the disease.

KEGG ID	Description	Gene Name
fca05134	Legionellosis	TLR4, CASP3, UBE2E2, LOC102967742, TRRAP, CNTN1
fca05145	Toxoplasmosis	LY96, TLR4, CASP3, SFXN1, MAPK8, MAP3K7, IFNGR1, ITGB1
fca05133	Pertussis	LY96, TLR4, CASP3, MAPK8, LOC102967742, ITGB1
fca05140	Leishmaniasis	SFXN1, MAP3K7, IFNGR1, TLR4, ITGB1
fca05414	Dilated cardiomyopathy	LOC102950759, SFXN1, CACNA1D, MYL3, TPM4, ITGB1
fca05144	Malaria	SFXN1, TLR4, HGF, GYPC
fca05014	Amyotrophic lateral sclerosis (ALS)	GRIA4, LOC102967742, CASP3
fca05016	Huntington’s disease	CLTCL1, LOC102962163, LOC102957675, CLTC, CASP3, NDUFA4, AP2A2, CREB3L2
fca05162	Measles	ADAR, TNFAIP3, IFNGR1, TLR4, MAP3K7
fca05168	Herpes simplex infection	TRAF5, CASP3, MAPK8, MAP3K7, SRSF6, IFNGR1, CCPG1
fca05164	Influenza A	ADAR, TMPRSS2, TLR4, MAPK8, LOC102967742, IFNGR1
fca05010	Alzheimer’s disease	YPEL2, ATP2A3, CACNA1D, LOC102953458, CASP3, NDUFA4
fca05012	Parkinson’s disease	CASP3, NDUFA4, GPR37, LOC102962163, UCHL1
fca05152	Tuberculosis	TLR4, CASP3, SFXN1, FARSB, MAPK8, IFNGR1
fca05146	Amoebiasis	SFXN1, TLR4, CASP3
fca05416	Viral myocarditis	CASP3

## Data Availability

Raw sequence data were submitted to the NCBI Sequence Read Archive (SRA) under accession number SRP156517 (SRX4514236, SRX4514235, SRX4514234, SRX4514233, SRX4514232, SRX4514231, SRX4514230, SRX4514229).

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
