# Peer review of "New Evidence of Tiger Subspecies Differentiation and Environmental Adaptation: Comparison of the Whole Genomes of the Amur Tiger and the South China Tiger"

_animals, 2022, doi:10.3390/ani12141817_

Round 1
Reviewer 1 Report
The paper is interesting however it needs some structural changes based on the following comments.
It is understandable that obtaining genetic material for such rare animals is not easy, however the number of tissues analyzed is very low. Therefore, I suggest to the authors to focus the paper exclusively on the identification of functional genes and regulatory pathways related to reproduction, disease, predation, metabolism and characterized functional genes related to survival in the wild in the two subspecies. The differentiation into two large genetic clades of the two subspecies that the authors found could be due to the origin of the specimens studied which come from highly reduced populations and therefore subject to bottlenecks. So these aspects together with the implications for conservation must be reduced and considered with caution.
Minor comments:
Please check the text carefully for many typos such as:
Line 92: change Puma Concolor Coryi with Puma concolor coryi
Line 235: change We with we
Line 437: cange Panthera with Panthera
Please clarify how many tissues are analyzed why in Line 107: “…. Seven tissue samples…” and in Line 111: “…. information of the eight samples…”
The figures are not very legible therefore they need to be improved
Author Response
It is understandable that obtaining genetic material for such rare animals is not easy, however the number of tissues analyzed is very low. Therefore, I suggest to the authors to focus the paper exclusively on the identification of functional genes and regulatory pathways related to reproduction, disease, predation, metabolism and characterized functional genes related to survival in the wild in the two subspecies. The differentiation into two large genetic clades of the two subspecies that the authors found could be due to the origin of the specimens studied which come from highly reduced populations and therefore subject to bottlenecks. So these aspects together with the implications for conservation must be reduced and considered with caution.
Response: Your advice is invaluable. Thank you very much. Currently, the Chinese government is actively promoting the reintroduction of south China tiger populations into the wild. We hope that genome-wide studies can help with the release of the South China tiger into the wild, so conservation is one clue for our research. Thank you very much for your valuable suggestions. We will make some modifications according to your suggestions.
Line 92: change Puma Concolor Coryi with Puma concolor coryi
Response: Thank you very much for your comments. We have revised it in the manuscript.
Line 235: change We with we
Response: Thank you very much for your comments. We have revised it in the manuscript.
Line 437: cange Panthera with Panthera
Response: Thank you very much for your comments. We have revised it in the manuscript.
Please clarify how many tissues are analyzed why in Line 107: “…. Seven tissue samples…” and in Line 111: “…. information of the eight samples…”
Response: We collected eight samples, including seven tissue samples and one blood sample, to accurately describe the nature of the sample, so different expressions.
The figures are not very legible therefore they need to be improved.
Response: We have uploaded the clear picture as an attachment.
Reviewer 2 Report
The manuscript “New evidence of tiger subspecies differentiation and environmental adaptation: comparison of the whole genomes of the Amur tiger and the South China tiger” describes the basic genetic characteristics of two endangered tiger subspecies using whole-genome resequencing. Overall, I feel the story is attractive but some of the conclusions are overstated without strong literature or data support. I recommend refining the writing, content, and scientific questions of the study to increase the novelty of the manuscript.
It seems the authors are simply reporting the data without curiosity. For instance, one of the important findings in this study is the shallow LD decay curvy in South China tiger compared to Amur tiger. The author attributes this difference to the inbreeding among limited South China tiger descendants. However, the LD estimates are strongly impacted by the number of individuals within each population. Besides, a direct measurement of effective population size or relatedness (or an interpretation based on different K results) may help test and explain the hypothesis. Therefore, I strongly recommend the authors revisit these analyses and further test their hypotheses.
The authors mentioned that this study can be the basis for the future conservation and breeding of tigers. How to define the conservation and evolutionary units from the genetic data? How to improve and preserve the genetic resource of these subspecies? Further discussions are needed in the manuscript.
Specific:
Line 56-57: “Currently, the world's wild individuals are about 3062 to 3948, and most of them were isolated in small populations” -- what tiger species are you inferring here? Any reference?
Line 59: change “Only” to “only”
Line 74-76: this sentence is logically non-sense. Please consider rephrasing the sentence.
Line 76-77: this sentence is overstated. The genome information may aid in the understanding of speciation and conservation units, but is not able to distinguish them. Please consider rephrasing the sentence.
Line 77: you’re trying to highlight the concepts of multiple conservation or evolutionary units. You need to provide the citation here.
Line 88-89: what is the “robust historical change data” here?
Line 90-92: to what extent will you consider a “successful case of species recovery”? Please provide the detailed criteria here.
Line 98: change “hoped to obtain” to “explored”
Line 136: Please explain how you obtained the average coverage of 99. 82%?
Line 175: change “used to detect” to “calculated for”
Line 221: Why the CNVs were identified based on coverage?
Figure 2: It is not necessary to list the LD parameters here.
Line 295: what criteria do you use as the cut-off for Fisher's exact test?
Line 309: What approach do you use for the multiple test correction?
Table 2: It seems that table 2 is incomplete, right?
Line 423-425: none of the analyses in this study is related to gene expression, need citation here.
Line 451-455: I don’t understand what the author is inferring here. Hard sweep through natural selection? Please consider rephrasing the sentence.
Line 469: change “significant” to “significantly higher”
Line 486-489: redundant contents, consider shortening the sentence.
Figures 1 and 2 are blurred and hard to follow. Please increase the resolution.
Throughout the manuscript, I did not see any link or supplementary content that related to the codes for data analyses. Please consider making these codes open for reproduction (this could be done after publication). It will act as an available reference for any ambiguity in the algorithmic descriptions of the manuscript.
Author Response
It seems the authors are simply reporting the data without curiosity. For instance, one of the important findings in this study is the shallow LD decay curvy in South China tiger compared to Amur tiger. The author attributes this difference to the inbreeding among limited South China tiger descendants. However, the LD estimates are strongly impacted by the number of individuals within each population. Besides, a direct measurement of effective population size or relatedness (or an interpretation based on different K results) may help test and explain the hypothesis. Therefore, I strongly recommend the authors revisit these analyses and further test their hypotheses.
Response: Thank you very much for your comments on the manuscript, which are very important for the revision of the manuscript. Your opinion is also of great help and reference value for future research.
LD of the South China tiger was significantly higher than that of the Amur tiger, which may be related to inbreeding and the effective population size of the South China tiger. About 66 years ago, the effective population of the South China tiger was lost. Inbreeding has taken place since captivity in 1963; Continuous inbreeding has also led to some problems that are not conducive to population reproduction; however, the inbreeding of Amur tigers in captivity is low. Principal component analysis and population structure analysis also showed that the South China tiger samples were closely related, while the Amur tiger samples were far related. Of course, the error caused by the small sample size cannot be ruled out, but this result should also be paid attention to by managers because closely related groups are likely to lead to inbreeding.
- Xu, Y.C.; Fang, S.G.; Li, Z.K. Sustainability of the South China Tiger: Implications of Inbreeding Depression and Introgression. Conserv. Genet. 2007, 8, 1199–1207.
- Zhang, L.; Lan, T.; Lin, C.; Fu, W.; Yuan, Y.; Lin, K.; Li, H.; Kumar Sahu, S.; Liu, Z.; Chen, D. Chromosome‐scale Genomes Reveal Genomic Consequences of Inbreeding in the South China Tiger: A Comparative Study with the Amur Tiger. Mol. Ecol. Resour. 2022.
-
Zhang, W.; Xu, X.; Yue, B.; Hou, R.; Xie, J.; Zou, Z.-T.; Han, Y.; Shen, F.; Zhang, L.; Xie, Z. Sorting out the Genetic Background of the Last Surviving South China Tigers. J. Hered. 2019, 110, 641–650.
- Liu, D.; Ma, Y.; Li, H.; Xu, Y.; Zhang, Y.; Dahmer, T.; Bai, S.; Wang, J. Simultaneous Polyandry and Heteropaternity in Tiger (Panthera Tigris Altaica): Implications for Conservation of Genetic Diversity in Captive Populations of Felids. Chin. Sci. Bull. 2013, 58, 2230–2236.
The authors mentioned that this study can be the basis for the future conservation and breeding of tigers. How to define the conservation and evolutionary units from the genetic data? How to improve and preserve the genetic resource of these subspecies? Further discussions are needed in the manuscript.
Response: The research shows that scientific management could reduce the risk of inbreeding. However, recent studies have thought that scientific management can only solve the current inbreeding and cannot solve long-term, so the scientists suggest that gene rescue can reduce South China tiger's inbreeding. Our study found genetic differences in environmental adaptation between the Amur tiger and the South China tiger. The Amur tiger is genetically adapted to the cold environment, which is contrary to the warm environment suitable for the South China tiger. These genes might become novel genetic loads for the South China tiger gene pool given they are introduced. We suggest that the recovery of the South China tiger population using gene rescue should be carefully selected for homologous species, especially for environmental adaptation.
- Xu, Y.C.; Fang, S.G.; Li, Z.K. Sustainability of the South China Tiger: Implications of Inbreeding Depression and Introgression. Conserv. Genet. 2007, 8, 1199–1207.
- Zhang, L.; Lan, T.; Lin, C.; Fu, W.; Yuan, Y.; Lin, K.; Li, H.; Kumar Sahu, S.; Liu, Z.; Chen, D. Chromosome‐scale Genomes Reveal Genomic Consequences of Inbreeding in the South China Tiger: A Comparative Study with the Amur Tiger. Mol. Ecol. Resour. 2022.
-
Zhang, W.; Xu, X.; Yue, B.; Hou, R.; Xie, J.; Zou, Z.-T.; Han, Y.; Shen, F.; Zhang, L.; Xie, Z. Sorting out the Genetic Background of the Last Surviving South China Tigers. J. Hered. 2019, 110, 641–650.
Specific:
Line 56-57: “Currently, the world's wild individuals are about 3062 to 3948, and most of them were isolated in small populations” -- what tiger species are you inferring here? Any reference?
Response: We used the IUCN database.
Goodrich, J.; Lynam, A.; Miquelle, D.; Wibisono, H.; Kawanishi, K.; Pattanavibool, A.; Htun, S.; Tempa, T.; Karki, J.; Jhala, Y.; et al. IUCN Red List of Threatened Species: Panthera Tigris. 2015.
Line 59: change “Only” to “only”
Response: Thank you very much for your comments on the manuscript. We have made modifications in the manuscript.
Line 74-76: this sentence is logically non-sense. Please consider rephrasing the sentence.
Response: Thank you very much for your comments on the manuscript. There may be some problems in language expression in this part, so we have modified the whole sentence in the manuscript.
Line 76-77: this sentence is overstated. The genome information may aid in the understanding of speciation and conservation units, but is not able to distinguish them. Please consider rephrasing the sentence.
Response: Thank you very much for your comments on the manuscript. There may be some problems in language expression in this part, so we have modified the whole sentence in the manuscript.
Line 77: you’re trying to highlight the concepts of multiple conservation or evolutionary units. You need to provide the citation here.
Response: Thank you very much for your comments on the manuscript. There may be some problems in language expression in this part, so we have modified the whole sentence in the manuscript. This section has been deleted.
Line 88-89: what is the “robust historical change data” here?
Response: Thank you very much for your comments on the manuscript. There is no relevant content in the research content, so this part is deleted.
Line 90-92: to what extent will you consider a “successful case of species recovery”? Please provide the detailed criteria here.
Your question is excellent. Thank you for your opinion. Currently, there is no standard for the recovery degree of the population, so we only refer to the recovery case of the Florida Panthers in the United States, but whether it applies to the South China tiger is unknown. So we cannot have a clear standard right now. I am terribly sorry.
Line 98: change “hoped to obtain” to “explored”
Response: Thank you very much for your comments on the manuscript. We have made modifications in the manuscript.
Line 136: Please explain how you obtained the average coverage of 99. 82%?
|
Sample |
clean reads |
mapped reads |
mapping rate |
Average depth(X) |
Coverage at least 1× |
|
Ha |
515,646,210 |
504,236,999 |
97.79% |
26.31 |
99.80% |
|
Hb |
501,949,856 |
490,761,477 |
97.77% |
25.16 |
99.81% |
|
Hc |
518,662,626 |
505,967,973 |
97.55% |
26.22 |
99.74% |
|
Da |
528,818,888 |
518,192,199 |
97.99% |
27.75 |
99.84% |
|
Db |
595,423,234 |
584,476,947 |
98.16% |
30.32 |
99.86% |
|
Dc |
608,819,436 |
596,136,727 |
97.92% |
30.67 |
99.85% |
|
Dd |
511,592,406 |
500,779,455 |
97.89% |
26.24 |
99.77% |
|
De |
535,663,888 |
524,170,250 |
97.85% |
27.15 |
99.85% |
Response: Thank you very much for your comments on the manuscript. As shown in the table, we have averaged the average coverage of 8 samples.
Line 175: change “used to detect” to “calculated for”
Response: Thank you very much for your comments on the manuscript. We have made modifications in the manuscript.
Line 221: Why the CNVs were identified based on coverage?
Response: I'm sorry, but this is a linguistic error.
” 268187 copy number variations (CNVs) were identified”
Figure 2: It is not necessary to list the LD parameters here.
Response: Thank you very much for your comments on the manuscript. We have made modifications in the manuscript.
Line 295: what criteria do you use as the cut-off for Fisher's exact test?
Response: P value<0.05 was used as the cut-off for Fisher's exact test
Line 309: What approach do you use for the multiple test correction?
Response: Bonferroni was used for the multiple test correction
Table 2: It seems that table 2 is incomplete, right?
Response: Because the table needs to reflect a lot of content, we put the specific content in supplementary table 14.
Line 423-425: none of the analyses in this study is related to gene expression, need citation here.
Response: Kousteni, S. FoxO1, the Transcriptional Chief of Staff of Energy Metabolism. Bone 2012, 50, 437–443.
Line 451-455: I don’t understand what the author is inferring here. Hard sweep through natural selection? Please consider rephrasing the sentence.
Response: Thank you very much for your valuable comments. After careful consideration, we have deleted this sentence.
Line 469: change “significant” to “significantly higher”
Response: Thank you very much for your comments on the manuscript. We have made modifications in the manuscript.
Line 486-489: redundant contents, consider shortening the sentence.
Response: Thank you very much for your comments on the manuscript. We have made modifications in the manuscript.
Figures 1 and 2 are blurred and hard to follow. Please increase the resolution.
Response: We have uploaded the clear picture as an attachment.

Reviewer 3 Report
Dear authors,
This work tries to study the differences in adaptive selection between Amur tigers and South China tigers by high-throughput sequencing technology. The manuscript is well written and structured, the introduction provides sufficient background, the cited references are relevant to research, the research design is appropriate, the methods are adequately described, the results are clearly presented, and the conclusions are supported by the results. Only minor formatting revisions are necessary. I recommend that authors review the format thoroughly.
Author Response
This work tries to study the differences in adaptive selection between Amur tigers and South China tigers by high-throughput sequencing technology. The manuscript is well written and structured, the introduction provides sufficient background, the cited references are relevant to research, the research design is appropriate, the methods are adequately described, the results are clearly presented, and the conclusions are supported by the results. Only minor formatting revisions are necessary. I recommend that authors review the format thoroughly.
Response: Thank you very much for your comments on the manuscript. We have revised the whole manuscript according to your comments.

Round 2
Reviewer 1 Report
The authors have partially followed the suggestions and overall the paper is now acceptable for publication
Reviewer 2 Report
The questions have been appropriately addressed by the authors. Thanks.